# Brain Investigation on Sexual Dimorphism in a Gynandromorph Moth

**DOI:** 10.3390/insects13030284

**Published:** 2022-03-14

**Authors:** Elena Ian, Xi Chu, Bente Gunnveig Berg

**Affiliations:** Department of Psychology, Norwegian University of Science and Technology (NTNU), 7491 Trondheim, Norway; xi.chu@ntnu.no (X.C.); bente.berg@ntnu.no (B.G.B.)

**Keywords:** gynandromorph moth, olfaction, antennal lobe, confocal images, digital reconstruction

## Abstract

**Simple Summary:**

The noctuid moth, *Helicoverpa armigera,* is one of the globally most damaging agricultural pest insects. Generally, exploration of the male- and female-specific neural architecture underlying its reproductive behavior is crucial for developing biological and environment-friendly alternatives to the traditional pest control management. In this study, we utilized the opportunity to uncover putative sex differences in *H. armigera* by comparing details in the brain anatomy between the male and female hemispheres in one gynandromorphic individual. The methods included synapsin immunostaining, confocal microscopy, and the digital reconstruction of several brain areas involved in processing input about odor and vision, respectively. The results demonstrated sex-specific arrangements applying to distinct olfactory neuropils, including not only the primary olfactory center, the antennal lobe, but also higher order levels involved in odor-associated memory formation.

**Abstract:**

The present study was dedicated to investigating the anatomical organization of distinct neuropils within the two brain hemispheres of a gynandromorphic moth of the species *Helicoverpa armigera.* High quality confocal imaging of a synapsin immuno-stained preparation combined with three-dimensional reconstructions made it possible to identify several brain structures involved in processing odor input and to measure their volumes in the male and female hemispheres. Thus, in addition to reconstructing the antennal lobes, we also made digital models of the mushroom body calyces, the pedunculus, and the vertical and medial lobes. As previously reported, prominent sexual dimorphism was demonstrated in the antennal lobes via the identification of a male-specific macroglomerular complex (MGC) and a female-specific complex (Fc) in each of the two brain hemispheres of the gynandromorph. Additionally, sex-specific differences were found in volume differences for three other neuropil structures—the calyces, pedunculus, and vertical lobe. The putative purpose of larger volumes of three mushroom body neuropils in females as compared to males is discussed.

## 1. Introduction

Dimorphic individuals, in which the bodies consist of both female and male parts, were observed before the term *gynandromorphism* was coined and described by Rudolfi for the moth species, *Gastropacha quercifolia,* in 1825. Though this phenomenon is more commonly reported in insects than in other animal classes, the estimated occurrence of insect gynandromorphs is about 0.01 to 0.05%. Gynandromorphism has been observed in several insect orders including Lepidoptera, Hymenoptera, Coleoptera, Diptera, and Orthoptera, but its occurrence across orders is not even [1]. Several thousands have been reported in Lepidoptera in comparison with 90 in Hymenoptera; however, this fact could be due to easily recognizable sexual dimorphic features in the former order [2].

Lepidoptera (moths and butterflies) have a female heterogametic sex chromosome system, with most females having a WZ constitution while males are ZZ [3]. The cause of bilateral gynandromorphism arises from an event in mitosis during a very early stage in the development of the organism. When the juvenile organism grows through a series of cell divisions, a genetic error in the chromosomes leads to one of the dividing cells splitting its sex chromosomes aberrantly, resulting in two dissimilar daughter cells, one having the chromosomes that cause female development and the other the chromosomes that cause male development.

While external dimorphism appears as morphological features like size, form, and color, internal dimorphism is reflected by gender-specific reproductive organs and brain anatomy. In our lab, the Chemosensory lab at the Norwegian University of Science and Technology (NTNU), an emerged moth of the species, *Helicoverpa armigera*, proved to be a bilateral gynandromorph, with a clear external segregation of the male and female appearances on the left and right side, respectively. In addition, internally observed bilateral brain structures gave us a unique opportunity to uncover putative sexual dimorphic traits of the moth brain, making a comparison of the gynandromorphs male and female hemispheres.

In the moth brain, the antennal lobe (AL) stands out as a prominent sexually dimorphic neuropil. Functionally, the moth AL, also called the primary olfactory center of the insect brain, plays a role in processing chemo-sensory information. The AL consists of spherical units, glomeruli, which are segregated into several groups based on their morphological and functional specificities [4]. In *H. armigera*, two recent studies performed a global-wide comparison between female and male ALs to identify sexually dimorphic and isomorphic glomeruli [5,6]. Three-dimensional reconstruction of the AL glomeruli revealed four distinct groups in both sexes. The most prominent sexual dimorphism in the moth species is demonstrated by the macroglomerular complex (MGC). In *H. armigera* males, there are three glomeruli grouped into the MGC and processing input about female-produced components [7]. In females, a small assembly of enlarged glomeruli, possibly being comparable with the MGC units in males, has been observed in several moth species: *Bombyx mori* [8], *Manduca sexta* [9], and *Heliothis virescens* [10]. In *H. armigera,* five enlarged glomeruli were identified as female-specific units and belonging to the female complex (Fx; [6]).

Generally, the largest group of glomeruli constitutes a category of so-called ordinary glomeruli (OG). In *H. armigera*, the OGs demonstrate high homology, though the numbers of glomeruli in males and females are different; in the *H. armigera* male there are 64–65 OGs and in female 66 [6]. The main task of these glomeruli is to process plant odor information. Additionally, there are also some reports about their involvement in thermal and mechanical stimuli processing as well [5,11].

Another category of glomeruli is the so-called posterior complex (PCx), a cluster of dorso-posteriorly located units. The function of this group is not yet defined, though it was shown in several male heliothine species that one of the PCx glomeruli receives input from two co-located distinct types of pheromone sensory neurons [7]. In addition, calcium imaging results also demonstrated pheromone-evoked responses in the PCx glomeruli (personal observations). The previous anatomical investigation of the AL in *H. armigera* males and females showed sexual dimorphism in PCx, including a different number of glomeruli: ten in males and nine in females [6]. The fourth glomerular category consists of a single large unit, the labial pit organ glomerulus (LPOG), receiving input from sensory neurons tuned to carbon dioxide [12]. The former investigations of *H. armigera*, showed that the LPOG volume is larger in females than in males [6].

Whether other neuropils than the primary olfactory center may also have a sex difference is still an open question. Thus, in the present study, we examined the *H. armigera* gynandromorph with respect to putative anatomical differences in the central brain areas of the two hemispheres. Brain regions including those for olfactory and visual processing were selected, as both olfactory and visual inputs are fundamental for moth navigation [13,14]. Comparative quantitative analysis of the digital neuropil reconstructions was performed on the optical lobes (OL), the anterior optic tubercle (AOTu), and the mushroom bodies (MB) including the calyces, lobes, and the pedunculus. The results demonstrate that the sex differences are not only present in the AL, but also in some of the protocerebral neuropils in the higher brain centers.

## 2. Materials and Methods

### 2.1. Insects

*Helicoverpa armigera* pupae (Lepidoptera; Noctuidae, Heliothinae), delivered by Keyun Bio-pesticides (Henan, China), were reared in climate chambers (Refritherm 200 and 6E, Struers-Kebolab, Albertsund, Denmark, or Binder KBF 720, Tuttlingen, Germany) at 24 °C and 70% air humidity on a 14:10 h light/dark cycle. The individual used in this study was one day old. According to Norwegian law of animal welfare, there are no restrictions regarding the experimental use of Lepidoptera.

### 2.2. Immunostaining

The preparation procedure was described in detail elsewhere [14,15]. In short, the brain was dissected in Ringer’s solution (in mM: 150 NaCl, 3 CaCl_2_, 3 KCl, 25 sucrose, and 10 N-tris (hydroxymethyl)-methyl-2-amino-ethanesulfonic acid, pH 6.9) and immediately transferred into a Zinc–Formaldehyde fixative [16] at room temperature overnight. The brain was then washed in HEPES-buffered saline (HBS, 8 × 30 min), and subjected to a permeabilization step (60 min incubation with a fresh mixture of 20% DMSO and 80% methanol) before being washed 3 × 10 min in a Tris-HCL buffer (0.1 M, pH 7.4). After pre-incubation in 5% normal goat serum (NGS, Sigma–Aldrich, St. Louis, MO, USA) in 0.1 M phosphate-buffered saline (PBS, pH 7.2) containing 0.3% Triton X-100 (PBT), the brain was incubated for 6 days at 4 °C in the primary antibody, SYNORF1 (dilution 1:25 in PBT containing 1% NGS). Following rinsing in PBT 8 × 30 min, the brain was incubated for 5 days at 4 °C with Alexa Flour 647 conjugated goat-anti-mouse secondary antibody solution (Invitrogen, Eugene, OR; dilution 1:300 in PBT with 1% NGS). After washing 4 × 30 min in PBT and 2 × 30 min in PBS, the brain was dehydrated in an increasing ethanol series (50%, 70%, 90%, 95%, and 100% (2×), 10 min each). Then, the brain was transferred to a mixture of methyl salicylate and ethanol (1:1) for 10 min and after that cleared completely in methyl salicylate for at least 1 h. Finally, the brain was mounted in a Permount mounting medium (Electron Microscopy Sciences) between two coverslips, separated by spacers.

### 2.3. Confocal Image Acquisition

The immunostained brain was imaged dorso-frontally by using a confocal laser scanning microscope (LSM 800 Zeiss, Jena, Germany) equipped with a Plan-Neofluar 20×/0.5 objective. The sample was excited with a HeNe laser at 653 nm and the fluorescent emission passed through a 650 nm long-pass filter at a voxel size of 0.62 × 0.62 × 2 μm. The confocal images shown in this study were edited in ZEN 2.3 (blue edition, Carl Zeiss Microscopy GmbH, Jana, Germany).

### 2.4. Digital Reconstruction and Volumetric Analysis

The high-quality confocal stack allowed for making 3D visualizations based on the auto-fluorescence signals without an immunostaining application. A complete image of the raw confocal stack of the gynandromorph moth was used for making digital reconstructions of the relevant brain structures by means of the AMIRA software (AMIRA 6.0, Visage Imaging, Fürth, Germany). The brain areas were manually demarcated by using a segmentation editor for all three spatial planes with a consequent wrap-tool to obtain full neuropil volumes. Surface models of the segmented areas were constructed, and their volumes were measured by the material statistics function in the segmentation editor.

### 2.5. Glomerular Naming

In correspondence with the previous classification of antennal-lobe glomeruli in the male and female heliothine moth, we grouped the units in four assemblies: (1) the male/female specific complex, (2) the ordinary glomeruli (OG), (3) the posterior complex (PCx), and (4) the labial pit organ glomerulus (LPOG) [5,6]. For the male specific MGC, the previously established names, Cu, Dmp and Dma were used. For the female counterpart, namely, the female complex, Fx, the three relevant glomeruli were named, Fx1–Fx3. Since the OGs in the male and female hemispheres could not be identified with respect to homologies, they were numbered without any correspondence. The OGs in the male part were named O_m_1–O_m_66 and in the female part O_f_1–O_f_66. For the PCx units, the names P_m_1–P_m_9 were used for the glomeruli in the male hemisphere and P_f_1–P_f_9 for the glomeruli in the female hemisphere. The LPOG unit was named L_m_ and L_f_ for male and female halves, respectively. In addition to the four above-mentioned glomerular assemblies, we created a new one, the ventroposterior complex (VPx), which was selected out of the OGs and named V_m_1–V_m_5 and V_f_1–V_f_5 in males and females, respectively.

### 2.6. Neuropil Volume Analyses

As gynandromorphic moths are rare, the analyses of the neuropile volume were conducted based on the comparison of data from one gynandromorphic brain and a male genotype representative brain. We used the hemispheric ratio to clarify the volumetric difference of corresponding neuropils between the two hemispheres. Thus, in the representative brain, the hemispheric ratio of a neuropil was computed by comparing the volume of corresponding neuropils in the left and right hemisphere. For the gynandromorphic brain, this ratio was calculated by comparing the neuropil volumes of corresponding structures in the male and female halves. For example, the hemispheric ratio of the vertical lobe in the gynandromorphic brain was calculated as the increase rate of vertical lobe volume in the female hemisphere relative to the male hemisphere. To determine whether the hemispheric ratios of neuropils in the gynandromorphic brain were different from those in the representative brain, we defined a threshold (T) based upon the hemispheric ratio of eight corresponding neuropils in the representative brain at a 10% significance level. We also compared the mean hemispheric ratios (across 8 neuropils) to a theoretically symmetrical brain, in which the hemispheric ratio should be 0, by using a one-sample *t* test. In addition, to investigate whether the hemispheric ratio was related with the neuropil function, we generated a hierarchical cluster analysis by using the centroid clustering method to obtain an overview of the linkage of all hemispheric ratios from, totally, 16 neuropils, including 8 in the representative brain and 8 in the gynandromorphic brain. All probabilities given are two-tailed. SPSS, version 25, was used for the statistical analysis.

## 3. Results

### 3.1. Anatomical Comparison between Female and Male Hemisphere in a Bilateral Gynandromorphic Brain

Digital reconstruction of the prominent and easily recognizable brain areas allowed a comparison of anatomical traits and volumes of the paired neuropils of two hemispheres in a bilateral gynandromorphic brain. We reconstructed six protocerebral neuropils in each hemisphere: the mushroom body calyces, the pedunculus, the vertical and medial lobes, the anterior optic tubercle and the central body. In addition, each antennal-lobe glomerulus and the three optical lobes, medulla, lobula, and lobula plate, were digitally labeled. Figure 1 visualizes the confocal sections and the detailed reconstruction of the labeled neuropils, where the male brain parts are presented in green shades and the corresponding female neuropils in magenta shades.

### 3.2. Proposed Dimorphic Neuropils in Heliothine Moths

The gynandromorphic sample with bilateral asymmetry provided a unique opportunity to compare morphological details of putative dimorphic brain structures between the female and male parts. We selected eight brain compartments with distinguishable boundaries, including three optical lobe neuropils (medulla, lobula and lobula plate) and five protocerebral neuropils (mushroom body calyces, pedunculus, vertical lobe, medial lobe, and anterior optic tubercle). The volume of each compartment was quantified based on two digital reconstruction datasets (Figure 2), one originating from the previously established representative brain [14], and the other from the bilateral gynandromorphic brain. The calculated hemispheric ratios were based on the volumes of paired neuropils within one reconstruction dataset. In the representative brain, the hemispheric ratio of a distinct neuropil was computed based on the volume of the neuropil in the left and right hemisphere, respectively. For the gynandromorphic brain, this ratio was calculated based on the neuropil volumes between the male and female halves. We defined a threshold (T) based upon the hemispheric ratio across eight neuropils in the representative brain at a 10% significance level (T = 10.73%). It was demonstrated that, in the representative male moth brain, none of the neuropil pairs had a hemispheric ratio above the threshold (Figure 2B,G); however, in the gynandromorphic brain, three neuropil pairs exceeded the hemispheric ratio threshold, i.e., the mushroom body calyces, vertical lobes, and pedunculi. In the female hemisphere, each of these three neuropils made up a larger volume than their counterparts in the male hemisphere (Figure 2D). The spatial organization of these three neuropil pairs are highlighted in Figure 2H.

Theoretically, in a wildtype moth brain, the volume of corresponding neuropils in the two hemispheres should be identical, which implies a hemispheric ratio of zero. To confirm the significance of the different volumes shown in the gynandromorphic brain, we compared the mean ratio of all eight neuropil pairs to 0. The one-sample *t* tests determined that the hemispheric ratios in the gynandromorphic brain were larger than 0 (*t* (7) = 2.97, *p* = 0.02), whereas the corresponding data in the representative brain were comparable with 0 (*t* (7) = 1.33, *p* = 0.23, Figure 2E). To further reveal the putative association between the neuropil pairs, we performed a hierarchical clustering test upon the hemispheric ratios in all 16 neuropil pairs, where 8 of them were collected from the male representative brain (R) and the other 8 pairs were from the gynandromorphic brain (G). Due to the restricted gynandromorphic brain sample size, the hemispheric ratio between each of the eight neuropil pairs in two brains was treated as an independent sample (N = 16). The test resulted in four clusters (Cluster I–IV, Figure 2F). Interestingly, the neuropil pairs in Clusters I and II, which demonstrated low hemispheric ratios, were mostly vision-associated brain areas. Almost all the neuropil pairs in Clusters III and IV, on the other hand, which had relatively high hemispheric ratios, were related to memory and learning.

### 3.3. Antennal Lobes: Glomerular Identification and Clustering

Comparison of the two ALs in the gynandromorphic brain clearly demonstrated a sexual dimorphism in the form of an assembly of sexually dimorphic glomeruli located at the antennal-nerve entrance. These sex-specific units made up the MGC and the Fx in the male and female hemisphere, respectively. As in a ‘normal’ brain, the additional AL glomeruli formed several segregated clusters forming seemingly sexually isomorphic glomeruli (Table 1). Generally, the male and female ALs of *H. armigera* are reported to consist of 67 glomeruli in both male and female parts. While the OGs, PCx, and LPOG are common categories previously described in both males and females, we classified a new cluster of glomeruli located ventro-posteriorly in the AL, the ventro-posterior complex (VPx). These glomeruli, positioned at the border between the AL and the protocerebrum, form a distinct group based on their atypical size, form, and compilation.

Three-dimensional reconstructions of the AL glomeruli allowed for making an overall comparison between the male and female halves; however, since it was not possible to define homologous glomeruli, the glomerular numbers on each side do not correspond to each other.

#### 3.3.1. Sex Specific Areas: MGC and Fx

In the bilateral gynandromorphic brain, the sex-specific AL glomeruli, the MGC and the Fx, were comparable to the corresponding glomerular structure in the normal male and female brain, respectively. The Fx in the gynandromorph sample, was identified as three enlarged glomeruli located at the area equivalent to the MGC (Figure 3).

#### 3.3.2. Ordinary Glomeruli

The ordinary glomeruli comprise the largest AL cluster. In the gynandromorph brain, there were 47 units in the male hemisphere and 50 in the female (Figure 4). In a previous study of the AL glomeruli in *H. armigera*, the majority of OGs in normal males and females were found to be homologous [6]. In the present investigation, it was not possible to perform such a detailed analysis. Therefore, the OG numbers for the two brain hemispheres listed here are not in correspondence. The total numbers of OGs identified in the ALs of the gynandromorph, 47/50, were reduced as compared to the quantity previously reported, 66/66 [5]. This is probably due to several factors, such as the classification of the new glomerular group, VPx, and the poorer staining quality of the glomerular units.

#### 3.3.3. Posterior Complex Glomeruli

Located posteriorly to the MGC/Fx in the gynandromorph, was the posterior complex (PCx) including 11 glomeruli in the male AL and 9 in the female (Figure 5). The number-labeling in the male and the female PCx does not correspond to homologous glomeruli. A prominent feature of the PCx is an enlarged glomerulus, number 1, located most dorsally. The calculated volume of this glomerulus in the male and female parts is notably different: 68.6 μm^3^ and 49.2 μm^3^, respectively.

#### 3.3.4. Labial Pit Organ Glomerulus (LPOG), and Ventroposterior Complex (VPx)

In this study, the glomeruli belonging to the VPx were described as a part of the ordinary glomeruli in the previous studies [5,6,17] and are presented here together with the labial pit organ glomerulus.

The LPOG was recognized in both the male and female AL of the gynandromorph. The volumes of the male and female LPOGs were distinct: 12.5 μm^3^ and 14.3 μm^3^, respectively. In the proximity to the LPOG, the ventrally located AL cluster, VPx, was localized. This glomerular assembly comprised four glomeruli in the female AL and five in the male (Figure 6). Among these enlarged glomeruli, V1 and V2 displayed a similar spatial location across the two ALs, indicating homology. The remaining VPx glomeruli in the male and female hemispheres seemed to be non-homologous.

## 4. Discussion

In the present study, we investigated the brain of a gynandromorphic *H. armigera* moth to uncover putative sex-specific neuropil regions. Comparison of the primary olfactory center, the AL, in normal males and females of this species has been conducted previously [5,6]; however, since neuropil volumes in different moth brains may depend on age, and pre- and post-eclosure experience, such comparisons across different individuals could reflect inter-individual variations rather than sex-specific distinctions. Thus, the gynandromorphic sample used in this study provided a remarkable opportunity to explore the putative sex differences in the same brain, where both the male and female hemispheres developed under the same conditions.

Confocal imaging and subsequent 3D reconstruction of individual brain regions in the gynandromorphic sample allowed us to make a precise volume comparison between several pairs of olfactory neuropils in the male and female hemispheres including the antennal lobe, calyces, pedunculus, and lobes of the mushroom bodies. In addition, we also examined the optical lobes regarding possible sex differences.

The most prominent sex-specific feature in the moth brain is the MGC of the male AL [5,18]. A previous study on a gynandromorphic moth of the species, *Agrotis ipsilon* (Hufnagel), demonstrated morphological differences within the brain and reproductive system [18]. Here, however, the ALs in both the male and female parts were incompletely developed in comparison with the typical male/female genotype. Thus, the AL of the *A. ipsilon’s* gynandromorphic brain included ordinary glomeruli in both hemispheres but no female-specific glomeruli in the female hemisphere and only a partially developed MGC in the male hemisphere. In our study on the gynandromorphic *H. armigera*, on the other hand, both ALs seemed to be completely developed. The anatomical organization in the two brain hemispheres was in full correspondence with the ‘normal’ male and female individuals, respectively. This included the presence of five glomerular assemblies in both hemispheres, the MGC/Fx, OGs, PCx, LPOG, and VCx.

As shown in the results, the arrangement of glomerular assemblies in the two brain halves of the gynandromorphic brain were comparable, with one exception—a sex-specific group formed by the MGC and Fx in the male and female hemispheres, respectively. This is in full agreement with previous data on this species reporting about a sex-specific group of glomeruli located at the antennal-nerve entrance into the AL [5,6,17].

Concerning the largest group of glomeruli, the OGs, comparison between the ALs of the gynandromorph *H. armigera* and a normal type showed a significant difference. In this study, there was a total number of 47 and 50 OGs in the male and female hemispheres, respectively; however, in the previous studies on *H. armigera*, [5,6], the respective numbers were 66 and 61 [5,6]. The relatively large difference between the numbers of identified glomeruli could be due to the different staining procedures applied. While only synapse-associated protein immunostaining was used in the present study, the former investigations combined multiple staining methods including two distinct immunostaining techniques, anterograde staining of the sensory axons, and retrograde staining of AL output neurons. The previous reports therefore allowed a staining quality probably visualizing distinct glomeruli in the deep parts of the AL-glomeruli that might not be recognized in this study. Moreover, the definition of the VCx glomeruli as a separate group in the present study, which were previously included in the OGs, leads to a lower number as well.

In addition to the anatomical difference in one part of the male and female AL of the gynandromorph, we found volumetric distinctions between the protocerebral neuropils in the two hemispheres. Altogether, the high resolution of the confocal images made it possible to reconstruct three optical lobe neuropils (medulla, lobula, and lobula plate) and five protocerebral neuropils (MB calyces, pedunculus, vertical lobe, medial lobes, and the AOTu). This contrasts with the fact that none of the corresponding neuropil pairs labeled in the representative brain had hemispheric ratios above the threshold. The three MB neuropil pairs exceeding the hemispheric ratio threshold in the gynandromorphic brain were the calyces, vertical lobe, and pedunculus.

Interestingly, the female hemisphere in the gynandromorph demonstrated enlarged volumes of neuropils within the MBs, a higher order region for olfactory memory formation. Like in other insects, the MBs of the moth are formed by several morphological types of Kenyon cells (KC) having distinct anatomical traits within the Ca and the MB lobes [19,20]. Notably, the KCs are assumed to play a role in adjusting odor representation according to experience [21]. The difference between the volumes of certain MB compartments as obtained in this study may indicate the involvement of distinct circuits for processing biologically relevant signals in males and females, respectively. While the male moth mostly orients for detecting and finding a suitable conspecific female, the female has to recognize suitable host plants for oviposition—a task involving appropriate behavior according to the ecological surroundings. This may be reflected in a bigger number of the KCs in the female, leading to the ability to identify, learn, and remember a large amount of potentially suitable plant odors. Therefore, the different volumes of the MB structures in the two hemispheres of the gynandromorph may be related to an increased number of KCs in the female part satisfying the special requirements during oviposition. While volumetric differences between the MB structures of male and female moths have not been previously reported, sex-specific projection patterns within the calyces are well documented. Thus, in males, projection neurons (PNs) originating from the MGC and the OGs are reported to terminate within distinct areas of the calyces whereas females seem to have no organized patterns of PNs within this neuropil structure [14,22,23].

In conclusion, the data from the gynandromorph moth investigated here, not only confirmed the previous findings about sexual dimorphism in the primary olfactory center of the moth brain, the AL, but also in the higher centers like the calyces, pedunculus, and vertical lobe of the mushroom body. It is worth noting that there is a limitation in our conclusion, in particular that the statistical analyses were conducted in only two individual brains, due to the rareness of a gynandromorphic brain; however, the difference between the male and female brains we revealed in the gynandromorphic brain sample is representative.

## Figures and Tables

**Figure 1 insects-13-00284-f001:**
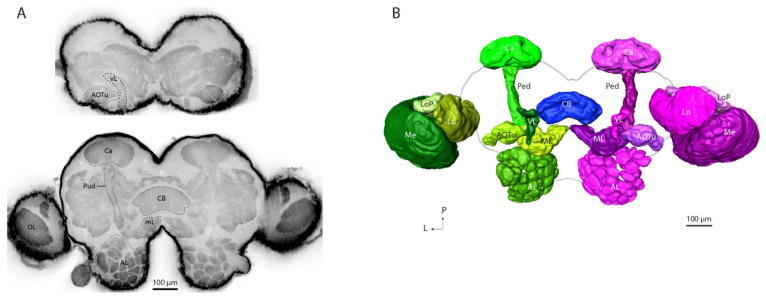
Anatomical organization of a bilateral gynandromorphic brain in *H. armigera*. (**A**) Confocal images of both hemispheres in the bilateral gynandromorphic brain. The two optical sections contain slices from a dorsal (depth: 57.98 µm, top image) to ventral (depth: 129.34 µm, bottom image). From the insect’s perspective, right hemisphere: male half; left hemisphere: female half. (**B**) Three-dimensional reconstruction (dorsal view) of selected neuropils in the gynandromorphic *H. armigera* brain. Each neuropil is in a shade of green/magenta, representing the male (green) and the female (magenta) hemispheres, respectively. AL: antennal lobe; AOTu: anterior optic tubercle; CB: central body; Ca: calyces; Lo: lobula; LoP: lobula plate; Me: medula; ML: medial lobe; Ped: pedunculus; VL: vertical lobe; OL: optical lobe.

**Figure 2 insects-13-00284-f002:**
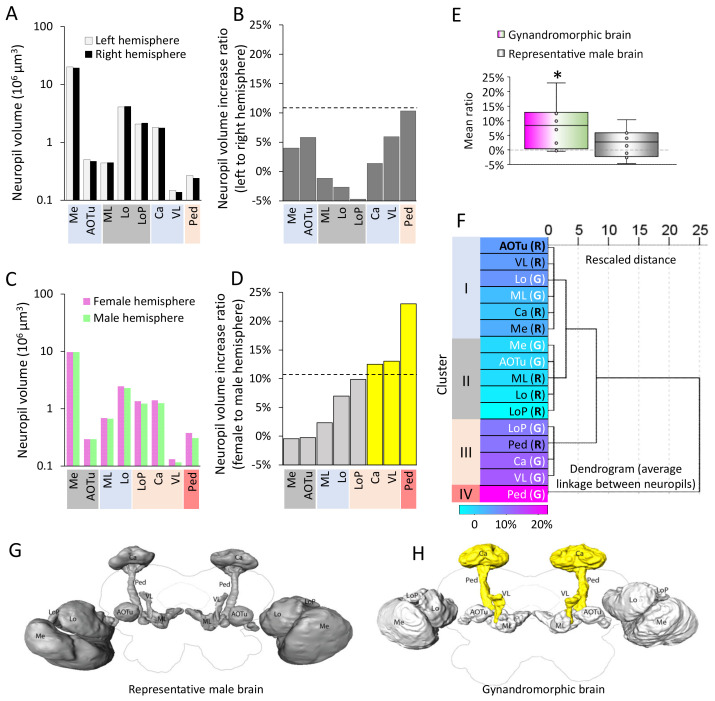
Comparison of distinct neuropil volumes in female and male hemispheres. (**A**) Volumes of 8 selected neuropils from both hemispheres in the representative male brain. (**B**) Volume ratios of these neuropils. The dashed line illustrates the hemispheric ratio threshold (T), computed from the hemispheric ratio in the representative brain at a 10% significance level. (**C**) Volumes of 8 selected neuropils from the female and male hemispheres in the bilateral gynandromorphic brain. (**D**) Hemispheric ratios of these structures. The dashed line illustrates the same kind of threshold (T) as shown in panel B. Columns in yellow show the 3 neuropils exceeding the threshold (T). (**E**) Comparison of the presumed hemispheric ratio (= 0, grey dash line) and the actual hemispheric ratio in the gynandromorphic and the representative brain, respectively. (**F**) Hierarchical clustering based on the hemispheric ratio of 16 neuropil pairs from the representative brain (indicated by R) and the gynandromorphic brain (indicated by G), including 8 left vs. right hemispheric neuropil pairs and 8 male vs. female hemispheric neuropil pairs (N = 16). Each neuropil was color-coded by the value of its hemispheric ratios. (**G**,**H**) 3D reconstructions of the 8 neuropiles in both brains. The neuropils having higher hemispheric ratios than the threshold (T) are marked in yellow. *, one-sample t test, compared with the hemispheric ratio in a theoretically symmetrical brain (0%), *p* < 0.05.

**Figure 3 insects-13-00284-f003:**
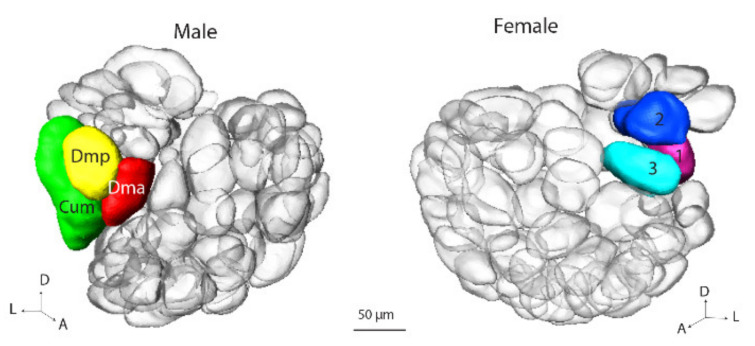
Three-dimensional reconstruction of the male macroglomerular complex (MGC) and the female complex (Fx). The MGC includes three units: the cumulus (Cum), the dorsomedial posterior (Dmp) and dorsomedial anterior (Dma) glomeruli. The Fx includes three enlarged glomeruli, having a similar location with the MGC.

**Figure 4 insects-13-00284-f004:**
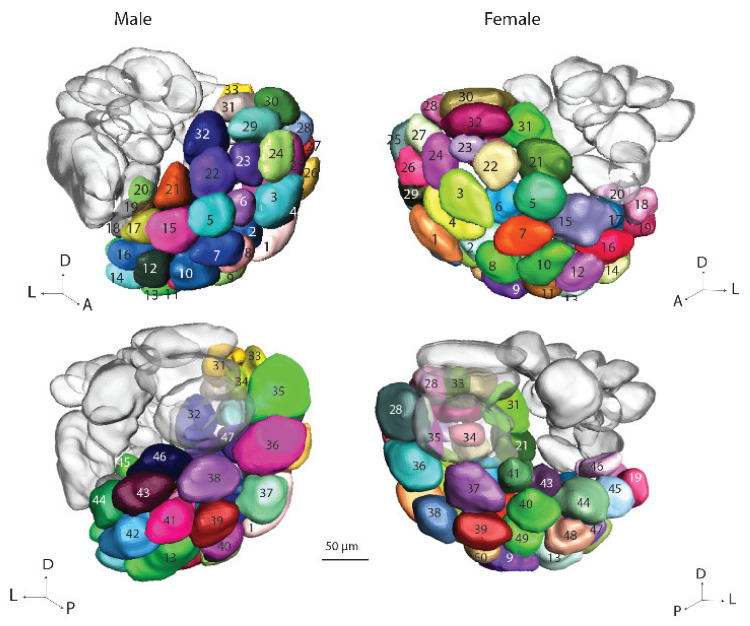
Three-dimensional reconstruction of the ordinary glomeruli in the male and female parts of the gynandromorph brain: dorsal view (**top**) and ventral view (**bottom**). The colors and numbers of the glomeruli in the two antennal lobes do not correspond to homologous units.

**Figure 5 insects-13-00284-f005:**
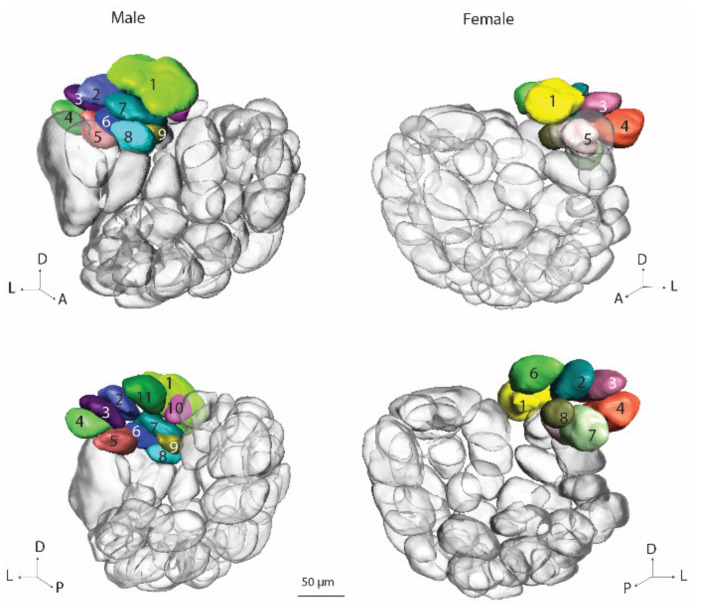
Three-dimensional reconstruction of the posterior complex in the male and female ALs of the gynandromorph: dorsal view (**top**) and ventral view (**bottom**).

**Figure 6 insects-13-00284-f006:**
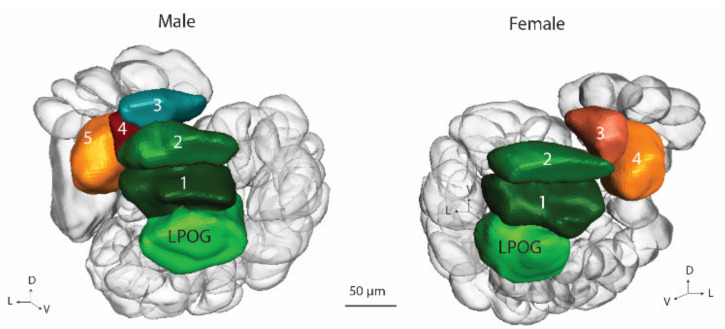
Three-dimensional reconstruction of the ventroposterior glomeruli (VPx) and the labial pit organ glomerulus (LPOG) in the male and female antennal lobe of the gynandromorph.

**Table 1 insects-13-00284-t001:** Categories and numbers of glomeruli in *H. armigera*, identified from the present and previous studies. MGC: macroglomerular complex; OG: ordinary glomeruli; PCx: posterior complex; VPx: ventroposterior complex; LPOG: labial pit organ glomerulus.

	Gynandromophic *H. armigera*		[5]		[17]	
	Male AL	Female AL	Male AL	Female AL	Male AL	Female AL
MGC/Fx	3	3	3	5	3	3
OG	47	50	66 ^a^	66 ^a^	61 ^b^	61 ^b^
PCx	11	9	10	9	-	-
VPx	5	4	-	-	-	-
LPOG	1	1	1	1	1	1
Total	67	67	80	81	65	65

^a^ the OG in [5] include the VPx. ^b^ the OG in [17]) include the PCx and VPx.

## Data Availability

Data sharing is not applicable to this article.

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
