# Peer review of "Brain Investigation on Sexual Dimorphism in a Gynandromorph Moth"

_insects, 2022, doi:10.3390/insects13030284_

Round 1

Reviewer 1 Report

The laboratory of Bente Berg has been studying the brain of Heliothine species for quite a while so there is no question that they are using the right methodology and that they know all the details of their system. It is a very nice paper, easy to read and to follow through.

My only concern regards to the statistical comparisons between the neuropils on the left and right brain hemispheres of a single individual. What they did "sounds" right but I am not 100% that it "is" right. If it weren´t it did not matter much because the paper is not based on those differences alone.

Very nice paper. Just some minor comments

  1. Abstract
    1. L24: prominent sexually dimorphism: prominent sexual dimorphism
    2. L26 found in the form of volume differences: found in volume differences
  2. M&M
    1. L122: 4x30 (in PBT: 4x30 min (in PBT
    2. L162: The term "rate". I think that "ratio" is more appropriate. Check throughout the document
    3. L169: the female hemisphere based on that in the male: the female hemisphere relative to the male
  3. Results
    1. L187: optic tubercle, respectively: I am not sure about the "respectively"
    2. Figure 1 caption: Where are the OL: optical lobes?
    3. Figure 2
      1. E: what is "*"?
      2. F: cluster analysis: Not clear
        1. Is it OK to do a cluster analysis with just 2 individuals?
        2. Not clear where the male, female, gynanfromoorph and norma male are
        3. What is G, What is R?
        4. L244: remove "totally"?
      3. L259: untypical: atypical
      4. Tanle 1: Footnote: "a" and "b" letters may not be necessary (L264,, 265)
      5. L282: 47 units in male hemisphere: 47 units in the male hemisphere
      6. Line 300: check the numbers of the volumes
      7. L303-304. Delete this phrase from Fig 5 caption: it is the heading of the next section
      8. L310: check numbers for the two volumes
    4. Discussion
      1. L353 and 355: check the number of glomeruli, they do not agree with Table 1
    5. References
      1. 12: Kent. Indicate journal

Author Response

Dear reviewer,

Thank you for your constructive suggestions. We have updated our manuscript according to them. 

The text was corrected according to all corresponding points. Here we include the answers only to some of them.

Figure 1 caption: Where are the OL: optical lobes?

Optical lobe is indicated on the confocal image.

Table 1: Footnote: "a" and "b" letters may not be necessary.

In our opinion, the footnotes are necessary to keep in order to indicate the papers to which corresponding brain areas were mentioned in.   

My only concern regards to the statistical comparisons between the neuropils on the left and right brain hemispheres of a single individual. What they did "sounds" right but I am not 100% that it "is" right. If it weren´t it did not matter much because the paper is not based on those differences alone.

The statistical analyses were first conducted on different neuropil pairs from an individual brain. We understand the reviewers concern, as the optimal comparisons shall be performed in a group of samples (e.g., N>6) to minimize error rates and improve the quality of hypothesis test. However, the rare occurrence of gynandromorphic individual makes the data collection of multiple samples less feasible in a short period. Thus, in order to perform the comparison, we had to compromise the sample size and treated the hemispheric ratio of each neuropil pair as an individual data point. Similar situation in the hierarchical clustering analysis, we included totally 16 neuropil pairs, 8 of them were collected from the male representative brain (R) and the other 8 pairs were from the gynandromorphic brain (G), and the hemispheric ratio between each of 8 neuropil pair in two brains was treated as an independent sample (N=16). This argument is now included in the revised manuscript.

Figure 2

E: what is "*"?

The indication of * is added in Line 251-253.

F: cluster analysis: Not clear

Is it OK to do a cluster analysis with just 2 individuals?

The explanation of clustering analysis is in Line 173-177 and Line 229-232.

Not clear where the male, female, gynanfromoorph and norma male are

What is G, What is R?

We have improved the figure legend of Fig. 2 according to the reviewer’s suggestion, see Line 247-248.

Reviewer 2 Report

The authors took advantage of a rare occurrence, the eclosion of a gynadromorphic moth, to study sexual dimorphism in the brain. Unlike comparisons between the brain of individual males and females, where slight variations in age and post eclosure experience could influence sexual differences, the gynadromorphic situation controls for such variables. The authors produced a simple study examining the major neuropils within the brain, identifying expected sexual dimorphism in some areas of the brain such as the MGC/Fx region. The female half also exhibited larger volumes of some neuropils associated with learning and memory, although the significance of this is as yet unknown.

The manuscript is well written, however there are some minor corrections to be made before it is ready for publication. Please see my comments below.

Line 24: correct to say “...sexual dimorphism”

Line 45- 49: revise sentence “When the juvenile organism grows through a series of cell divisions, a genetic error in the chromosomes leads to one of the dividing ...”

Line 75: remove the ‘a’, “...the OGs demonstrate high homology…”

Line 122: time unit missing, 4 x30 min

Line 204: boundary should be changed to boundaries

Line 255: “...AL’s of H. armigera are reported...”

Line 255: Table 1 correctly referred in the body text however it is labelled Table 2 in the table title.

Line 256: The body text and table incorrectly state that the female glomeruli number is 66, if you count up the various neuropil classes in the table the female also has 67. Please check this number.

Line 276. Figure legend should be “Three dimensional...” Note this is also wrong for figures 4, 5 and 6.

Line 277: the figure is missing the label for the cumulus (Cu) in the MGC.

Line 288: The reference for the 66/66 should be [5], not 17.

Line 300: the units should be μm3

Line 304: the LPOG and VPx are not labelled on the figure 5.

Line 310: the numbers need spaces and the unit should again be μm3

Author Response

Dear reviewer,

Thank you for your constructive suggestions. We have accepted all of them and the text was improved in correspondence with all of the points.